

# The dilemma in identifying WMO-defined tropopause height using high-resolution radiosondes

Yu Gou[1]; Jian Zhang[1*]; Wuke Wang[2]; Kaiming Huang[3]; Shaodong Zhang[3]

[1]Hubei Subsurface Multi–scale Imaging Key Laboratory, School of Geophysics and Geomatics, China University of Geosciences, Wuhan 430074, China

[2]School of environmental studies, China University of Geosciences, Wuhan 430074, China

[3]School of Electronic Information, Wuhan University, Wuhan 430072, China

*Correspondence to:* Dr. Jian Zhang (Email: zhangjian@cug.edu.cn)

**Abstract.** The tropopause plays a critical role in stratosphere–troposphere exchange and climate change. Its height is conventionally defined based on the World Meteorological Organization (WMO) threshold temperature gradient, yet this gradient is intrinsically linked to vertical resolution. Data with higher vertical resolution inevitably reveal finer gradient structures. While in situ radiosonde temperature measurements are considered the most reliable source for tropopause structure, high-resolution (5–10 m) soundings would be expected to yield more precise height estimates. The near-global coverage of high-resolution radiosondes, accumulated over even decades, promises valuable insights into long-term tropopause variability. However, our analysis demonstrates that the original WMO definition can lead to an underestimation of the tropopause height when using high-resolution soundings, potentially misidentifying the tropopause within thin inversions or temperature gradient discontinuities below tropopause. To address this, we leverage ERA5 tropopause heights as a reference to develop a high-resolution-optimized method. We evaluate three methods: original WMO method, Moving average method, and Coarse–Fine method. The results reveal that the mean differences between the three methods and ERA5 were 800 m, 280 m, and 180 m, respectively. Notably, ERA5 systematically overestimated the tropopause height compared to all methods, with this discrepancy particularly pronounced in the edges of the Hadley circulation. The proposed Coarse–Fine method, by effectively bypassing thin inversions and gradient extrema while preserving the fine–scale structure of the tropopause height, presents a promising tool for future investigations into long-term tropopause trends.



## Introduction

The tropopause, marking the boundary between the turbulent troposphere and the stably stratified stratosphere, is a crucial region for exchange of energy, air masses and water vapor (Xian and Homeyer, 2019). Furthermore, given the impact of global warming and ozone depletion on the troposphere and stratosphere, tropopause variations can serve as an indicator of anthropogenic environmental influences (Santer et al., 2003). Moreover, its extreme sensitivity to climate variability and change makes the tropopause a pivotal factor in understanding and predicting future climate scenarios (Sausen and Santer, 2003; Seidel and Randel, 2006).

The tropopause exhibits unique chemical and dynamical characteristics, and its maintenance relies on complex interactions between large-scale and small-scale circulation patterns, deep convection, cloud formation, and radiation (Randel and Jensen, 2013). For instance, water vapor abundance can influence tropopause height, as an increase in tropopause height accompanies increased optical thickness to maintain a constant emitted temperature. In addition, tropopause height is generally controlled by a combination of diabatic forcing and adiabatic dynamical effects (Zurita–Gotor and Vallis, 2013). These intricate mechanisms eventually lead to a marked difference in tropopause height between the tropics and the poles, with the annual average reaching approximately 16 km in the tropics and 8 km in the polar regions. The transition across the Hadley cell edge, near the subtropical jet, is particularly abrupt (Hu S. and Vallis, 2019).

However, defining the tropopause lacks a universal approach, relying instead on several empirical criteria based on properties exhibiting sharp transitions between the troposphere and stratosphere. While various definitions exist (e.g., cold point tropopause (CPT), dynamic tropopause), each presents limitations. The CPT is unsuitable for extratropical regions (Highwood and Hoskins, 1998), and the dynamic tropopause fails in part of mid-latitude regions (Thuburn and Craig, 1997). In contrast, the thermodynamic tropopause definition proposed by the World Meteorological Organization (WMO, 1957) offers a more robust global approach, providing reliable tropopause height estimates from various datasets (Hoffmann and Spang, 2022). Besides, increasing evidence suggests a recent upward trend in tropopause height (Santer et al., 2003; Sausen and Santer, 2003; Seidel and Randel, 2006). For instance, Seidel and Randel (2006) observed a $64\pm21$ m/decade upward trend from 1980 to 2004, and Son et al. (2009) projected continued future increase, albeit with a weaker trend. Analyses by Xian and Homeyer (2019) using radiosonde observations and reanalysis datasets from 1981–2015 detected a significant upward trend (40–120 m/decade). More recently, Zou et al. (2023) leveraged ERA5 (European Centre for Medium–Range Weather Forecasts (ECMWF) Reanalysis v5) data to reveal a widespread upward and cooling trend in the tropical tropopause from 1980–2021, demonstrating an increase of approximately $60 \pm 10$ m/decade (95% confidence). It is worthwhile to note that these radiosonde-related studies use data from the Integrated Global Radiosonde Archive (IGRA), a global dataset with coarse vertical resolution (approximately 300 to 400 m), and the ERA5 137-level model, however, has a vertical resolution of roughly 300 m at altitudes of 5 km and above, both these radiosonde data and reanalysis/model datasets, while useful for studying global tropopause variations, suffer from limitations in vertical resolutions (Raman and Chen, 2014).

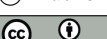



Instead, the high vertical resolution (5–10 m) of the radiosonde data allows for fine detailed observation of temperature structure changes within the troposphere and stratosphere. High-resolution sounding data enable the detection of fine-scale

tropopause structures over hundreds of stations. Furthermore, the global dataset is accumulating, with increasingly long records approaching climate-relevant timescales. Examples include the US (starting in 2005; Ko et al., 2019), China (starting in 2011), and Europe (starting in approximately 1991). These datasets hold great promise for advancing climate research and producing more refined and reliable long-term trend analyses in terms of tropopause structures, though current research in this area remains limited. It is worth noting that high-resolved radiosonde can improve the understanding of the tropospheric

processes, particularly tropospheric inversions (Fujiwara et al., 2003; Zhang et al., 2009). In principle, tropopause also exits as inversions. The WMO (1957) definition outlines a method for calculating the lapse-rate tropopause from temperature profiles, but lacks explicit justification for the specific lapse rate thresholds (e.g., 2 K/km) and depth criteria (e.g., 2 km, 1 km; Hoffmann et al., 2022). Frierson (2008) adopted a slightly different definition (4 K/km instead of 2 K/km) due to limitations inherent in the latter. Levine and Schneider (2015) found that the WMO definition struggles with simulations

involving near-neutral convection loss rates, thus questioning its applicability with high-resolution sounding data.

By contract, reanalysis datasets offer a globally continuous spatial and temporal representation of tropopause height, complementing radiosonde data. Datasets like ERA-Interim are widely recognized for their utility in climate monitoring (Dee et al., 2011), overcoming the spatial resolution limitations of radiosonde data. ERA5 tropopause height, the successor to ERA-Interim, has naturally been subjected to comparisons and evaluations with radiosonde in different regions (e.g., Zhu

et al., 2021; Velikou et al., 2022; Hoffmann et al., 2022). However, global comparative analyses of high-resolution (10 m) radiosonde data with ERA5 at the tropopause level remain scarce.

Consequently, this study addresses the following questions: (1) Is the directly WMO-defined method applicable to high-resolution soundings? If not, what alternative methods can improve its performance? (2) How well does ERA5 compare with high-resolution soundings in terms of tropopause height? To this end, section 2 details the data and the definitions and

limitation of tropopause retrieve methods used. Section 3 presents the distribution and comparison of radiosonde and ERA5 tropopause heights. Section 4 ends with a short summary.

## 2. Data and method

### 2.1 Radiosonde

Radiosondes are fundamental and crucial data sources for numerical weather prediction models. They are typically

carried aloft by weather balloons and burst at altitudes of approximately 27 km (Kumar, 2023). As the radiosonde ascends, it transmits meteorological data including temperature, pressure, relative humidity and air pressure to ground, sea, or air-based receiving stations (Durre et al., 2006). Globally, radiosondes are launched at approximately 800 sites regularly at twice a day (Durre et al., 2018; Ingleby et al., 2016). Radiosonde data are widely used in studies of planetary boundary layer height (Seidel et al., 2010; Sorbjan and Balsley, 2008), tropopause structure (Birner, 2006; Seidel and Randel, 2006; Sunilkumar et





al., 2017) and gravity waves (Ki and Chun, 2010; Yoo et al., 2018). Radiosonde data possess advantages derived from their in situ measurement characteristics, high vertical resolution, and generally reliable data sets. However, limitations include relatively low and uneven spatial resolution and a typically limited temporal frequency of typically twice daily measurements.

The data used in this study were sourced from institutions such as the University of Wyoming, ECMWF, and NOAA
(Shao et al., 2023). The data cover the year 2022, with a vertical resolution of 5–10 meters, ultimately sampled at 10 meters. Data acquisition was typically performed at 00 UTC and 12 UTC each day. To ensure data quality, only radiosonde stations with a minimum of 60 records conforming to the WMO definition of the first tropopause in 2022 were selected. This resulted in a dataset from 387 stations, comprising a total of 151,474 vertical profiles.

## 2.2 WMO-defined tropopause

Several methods exist for defining the tropopause. The CPT is reliable primarily in the tropics (between 20° S and 20° N) (Xian and Homeyer, 2019). The dynamic tropopause, based on Rossby-Ertel potential vorticity (PV), is less reliable in regions of low absolute vorticity, such as the tropics, and sometimes in mid-latitudes where strong anticyclonic flow prevails (Hoerling et al., 1991). In contrast, the WMO tropopause definition is more robust and generally applicable across a wider range of latitudes. Therefore, calculating tropopause height from radiosonde data using the WMO definition is currently the
most suitable method for global tropopause comparisons. As provided by the WMO:

(a) The "first tropopause" is defined as the lowest level at which the lapse rate decreases to 2°C/km or less, provided also the average lapse rate between this level and all higher levels within 2 km does not exceed 2°C/km;

(b) if above the first tropopause the average lapse rate between any level and all higher levels within 1 km exceed 3°C/km, then a "second tropopause" is defined by the same criterion as under (a). This tropopause may be either within or above the 1
km layer.

The lapse rates are calculated as follows:

$$\Gamma(z_i) = -\frac{\delta T}{\delta z} = -\frac{T_{i+1} - T_{i-1}}{z_{i+1} - z_{i-1}} \qquad (1)$$

with $T$ represents temperature, and $z$ represents geopotential height.

ERA5, the latest fifth-generation global atmospheric reanalysis product, stands out as one of the best high–resolution
atmospheric reanalysis products currently available, utilizing the ECMWF Integrated Forecasting System (IFS) Cy41r2, combined with a 4D-Var assimilation scheme. This remarkable initiative within the Copernicus Climate Change Service (Thépaut et al., 2018) benefits from advancements in modeling and data assimilation over a decade, providing a comprehensive and high-quality record of essential climate variables (Raoult et al., 2017).

Hoffmann et al. (2022) employed the WMO definition to calculate global tropopause heights from ERA5 data covering
the period 2000–2022, making this ERA5-based product available for research purposes. This study utilizes the 2022





ERA5-based tropopause height data, characterized by a horizontal resolution of 0.3° × 0.3° and a temporal resolution of 1 hour.

## 2.3 Moving average and Coarse–Fine methods

Directly applying the WMO definition to high-resolution radiosonde data, a method we call WMO direct method
(WDM), does not always effectively identify the tropopause height. Figure 1 (a) and (b) illustrate this issue, showing that WDM applied to high-vertical resolution radiosonde data may fail to accurately detect the tropopause structure. The presence of thin inversion layers can interfere with the algorithm. Similar issues occur in higher latitude stations as depicted in Figure 1 (c) and (d). Previous studies have observed inversions at lower altitudes in the troposphere (e.g., Fujiwara et al., 2003; Zhang et al., 2009), indicating that this phenomenon is not a random occurrence. To address these limitations, this
study proposes two new methods:

(a) Moving average method (MV): This method involves applying a moving window average to the radiosonde data (without altering the data resolution), followed by tropopause height detection using the WMO definition.

(b) Coarse–Fine method (C-F): This method first down samples the radiosonde data (changing the resolution according to the sampling window). The WMO definition is then applied to the down sampled data, resulting in a first tropopause height
detection. A second detection is then conducted using the initial high-resolution data within a specified range (defined by the size of two sampling windows) around the first detection, resulting in the final tropopause height.

We preliminarily investigated using a 500 m sliding window (SW) and a 500 m down-sampling interval (DI) within the MV and C-F, respectively. As shown in Figure 1, MV and C-F effectively circumvent the interference of thin inversions at lower altitudes in identifying the tropopause height. Figure 1 (e) and 1 (f), representing stations in low latitudes, demonstrate
this consistency. As discussed in Section 2.2, the CPT is considered reliable in tropics. The results obtained using MV and C-F are largely consistent with the tropopause heights determined using the cold point method in this region.

The two proposed methods address the issue of thin inversions in different ways: MV filters out the thin inversions by smoothing the data, while C-F ignores them by adjusting the sampling interval. While the tropopause itself is also an inversion structure (but a larger one) in the temperature-height profile, the increasing resolution of observational data allows
the WDM to potentially detect these lower-altitude "first tropopauses". Considering Figure 1, the tropopause heights obtained using MV and C-F are likely to be more consistent with previous research and understanding of the tropopause than those derived directly using the WMO definition.

## 3. Result

### 3.1 Sensitive analysis

Although the 500 m approach successfully avoided thin, low-altitude inversions, further analysis by using different SWs or DIs is needed to more accurately identify the tropopause. Considering the varying tropopause heights and activity





levels across different latitudes, a sensitivity analysis was performed using radiosonde stations in high, middle, and equatorial latitudes. Figure 2 presents monthly average analyses for five stations in 2022, using a series of SWs and DIs (60 m, 100 m, 300 m, 500 m, 700 m, 900 m). The left panels (a–e) depict the results using MV. A consistent upward

enhancement in average tropopause height is observed as the SW size increases from the WDM. In these plots, a roughly linear increase in calculated tropopause height is evident, with the 900 m SW average approximately 822 m higher than the WDM. The right panels (f–j) show the results using C-F, where the tropopause height variations form a roughly arched pattern. Peak values are typically observed at DI of 300 m or 700 m. The average peak height for the five stations in these plots is approximately 810 m higher than the WDM. In all ten subfigures, both methods consistently yielded tropopause

heights significantly higher than those determined by the WDM.

Prior to latitudinal analysis, both ECMWF model and radiosonde data were divided into seven climate zones: Northern Hemisphere/Southern Hemisphere polar (70°–90°), Northern Hemisphere/Southern Hemisphere mid-latitude (40°–70°), Northern Hemisphere/Southern Hemisphere subtropics (20°–40°), and tropics (20° S–20° N) (Houchi et al., 2010). Statistical analysis was performed on the station data within each latitudinal band. Given the limited number of station data available

between 70° S and 90° S in Antarctica, this region is excluded from the present analysis. As shown in Figure 3, it demonstrates that the temporal variation of tropopause height determined by MV, C-F and WDM is generally consistent. However, both MV and C-F exhibit a consistent enhancement of the tropopause height compared to the WDM. Those findings reinforce the findings presented in Figure 1. More specifically, Figure 3 shows that MV overestimates the tropopause by approximately 530 m (SW 300 m) and 700 m (SW 700 m) relative to the WDM. Similarly, C-F results in a

higher tropopause altitude by approximately 650 m (DI equals 300 m) and 630 m (DI equals 700 m).

Given the overestimations observed with both proposed methods compared to the WDM, a detailed sensitivity analysis was conducted by averaging data across the six latitudinal bands. Figure 4 and Table 1 reveal that both MV and C-F result in higher tropopause heights than WDM. Furthermore, this overestimation generally increases with larger SW and DI, ranging from 180 m to 800 m when SW and DI increasing from 60 m to 900 m. Interestingly, across the six latitude bands, the

tropopause heights obtained using different SWs or DIs within each method exhibit strong similarities in terms of temporal variation, aligning with the results observed in Figure 2. These findings corroborate the earlier hypothesis that WDM may potentially underestimate the true tropopause height due to the detection of lower-altitude "first tropopauses."

To leverage the high resolution of the radiosonde data while minimizing information loss and the influence of thin inversions, we chose parameters that aligned with the vertical resolution of the ERA5-based product. Thus, we prioritize C-F

with DI of 300 m and MV with SW of 300 m. Consequently, we select WDM, MV (SW equals 300 m), C-F (DI equals 300 m) for comparison with the ERA5 tropopause heights to validate these methods and their results.

## 3.2 Overall comparison between radiosonde and ERA5 defined tropopause height

Figure 5 presents the global distribution of annual average tropopause height derived from the ERA5 reanalysis in 2022, alongside a comparison with radiosonde data from individual stations. The figure reveals a distinct latitudinal gradient, with





tropopause heights notably lower at higher latitudes compared to mid-latitudes and low-latitudes. A marked transition in tropopause height occurs around 30° in latitude, exhibiting near-symmetrical patterns in both hemispheres. Comparison between the ERA5-based product and tropopause heights determined by the three different methods (WDM, MV and C-F) may reveal significant discrepancies, particularly evident in the Middle East where the radiosonde observation is sparse, with differences of around 2 km. Despite these differences, all three methods demonstrate a high correlation coefficient (>0.97) with the ERA5-based product in terms of annual average tropopause height, indicating a strong overall agreement.

Furthermore, Figure 6(a) reveals a substantial discrepancy between the ERA5 tropopause heights and those determined by the radiosonde. Approximately 90% (347/387) of stations exhibit a difference of 500 m in the absolute value between ERA5 and WDM. This substantial difference suggests a significant systematic bias. In contrast, Figure 6(b) and 6(c) demonstrate less pronounced differences between the ERA5-based product and the two alternative radiosonde methods. Only 22% (85/386) and 25% (97/386) of stations, respectively, exhibit an absolute difference of 500 m for MV and C-F. These results indicate that the ERA5 reanalysis overestimates tropopause height in the majority of cases compared to all three-defined radiosonde methods.

### 3.3 A detailed assessment of the four tropopause height datasets

Figure 7 displays seasonal comparisons of latitude-averaged tropopause heights between the ERA5-based product and three radiosonde methods (WDM, MV and C-F) in 2022. A notable seasonal pattern emerges: during spring and winter, tropopause height decreases significantly in the boreal latitude band of 20°–40°, while the decrease is less pronounced in the austral with equivalent latitude band. In addition, the figure also highlights the largest discrepancies between the ERA5-based product and the tropopause heights calculated by the WDM, followed by MV and the smallest discrepancies are found with C-F.

Statistically, Table 2 indicates that C-F outperforms the other two methods in terms of correlation coefficient and Centered Root Mean Squared Error (CRMSE). Although the standard deviations of the three methods exhibit relatively small differences, C-F displays a marginally higher standard deviation (approximately 0.05 km) compared to the other two methods.

The density distribution of tropopause heights, depicted in Figure 8, exhibits a bimodal pattern with two distinct peaks, one centered around 11 km and the other around 16 km. These peaks correspond to the tropopause heights in mid-high latitudes and equatorial regions, respectively, with a trough located around 13 km. This bimodal distribution with a minimum near 13 km pressure altitude separating tropical and extratropical modes aligns with the findings of Homeyer and Bowman (2013). More specifically, the ERA5-based product shows a wider range of tropopause heights (larger variance) compared to the three radiosonde methods. However, the lower quartile and median values of the ERA5-based product are closely aligned with those of MV and C-F. The upper quartile of the ERA5-based product is relatively higher. These results highlight that the ERA5-based product, compared to MV and C-F, exhibits a narrower waist and wider head in the tropopause height



distribution. This implies that the ERA5-based product likely overestimates tropopause height in the regions flanking the Hadley circulation.

## 4. Conclusion and discussion

Tropopause height is an indispensable metric in climate change research, directly influencing the temperature structure of the troposphere and stratosphere. Direct observations of the tropopause via radiosonde are generally most reliable, but are limited by the uneven global distribution of measurement stations, posing a significant constraint on global climate change studies. The temporally and spatially continuous ERA5 reanalysis dataset is now widely utilized in global climate analysis. Therefore, analyzing the differences between high vertical resolution radiosonde-derived tropopause data and the ERA5

reanalysis dataset is crucial.

This study investigates the spatial and temporal discrepancies between ERA5-derived tropopause heights and near-global high-resolution radiosonde data for 2022. Building upon previous research by Hoffmann and Spang (2022), our investigation confirms a persistent systematic overestimation of tropopause height in ERA5 reanalysis data compared to radiosonde despite their use of 25–30 m vertical resolution compared to our 10 m resolution. Furthermore, our analysis

reveals that the thermodynamic tropopause, as defined by the WMO, may exhibit a similar characteristic to thin, low-altitude inversions observed in high-resolution radiosonde data. This leads to an artificially low thermodynamic tropopause value. Consequently, we argue that directly applying the WMO definition to calculate the tropopause from high-resolution radiosonde data is not necessarily straightforward or reliable.

In order to address the issue of artificially low tropopause heights detected by WDM, we implemented two alternative

approaches, namely the moving average method (MV) and the coarse-fine method (C-F), and subsequently performed a sensitivity analysis. Guided by the vertical resolution of the ERA5 137-level model (approximately 300 m), we selected a 300 m sliding window or down-sampling interval for MV or C-F, respectively, while retaining the WDM for comparison. Our analysis indicates that both MV and C-F result in a more spatially and temporally consistent elevation of tropopause height compared to the WDM. Specifically, MV and C-F yielded approximately 500 m and 600 m higher tropopause heights,

respectively. This suggests that the systematic downward bias in tropopause heights derived from WDM is a widespread phenomenon. In our analysis, we tend to use C-F, as it effectively preserves the fine structural details of the tropopause while omitting the thin inversion layer. This approach ensures a more accurate representation of the tropopause's intricate features without compromising the overall integrity of the data.

The ERA5-based product exhibits a tendency to overestimate tropopause height in transition zones (e.g., the edges of

the Hadley circulation) compared to all three radiosonde methods. Despite median and lower quartile values that are largely consistent with MV and C-F, globally, the ERA5 tropopause heights remain approximately 280 m and 180 m higher than these methods, respectively. The overestimation is even greater, roughly 800 m, when compared to the WDM. These results suggest that the WDM is susceptible to the influence of thin inversion layers, leading to an underestimation of tropopause



height, while ERA5-based product estimates tend to overestimate tropopause height in transition zones compared to MV and
C-F.

However, the impact of window size selection remains an uncertainty, insufficient window sizes are unable to resolve thin inversion layers and sharp temperature gradients, while overly large windows may misrepresent the true tropopause height. To enhance the determination of tropopause height, it may be advantageous to integrate multi-parameters, including surface temperature, tropospheric water vapor content, upper-level temperature profiles, and historical information of
tropopause height. Advanced methodologies, such as deep-learning-based time-series prediction and inertial algorithms, could be employed to analyze parameters and predict the most likely tropopause height. This integrated approach might provide a more robust and accurate identification of tropopause position, offering valuable insights for future research in this domain.

In summary, these findings suggest that the WMO's definition, when applied to high-resolution atmospheric data, may
systematically fail to capture the thin, discontinuous temperature inversions within the troposphere. Specifically, our analysis reveals that directly applying the WMO criteria to data with fine vertical resolution (e.g., 10m) necessitates implementation of modified methodologies, such as the MV and C-F approaches proposed herein, to mitigate biases. Furthermore, tropopause heights from ERA5 reanalysis are consistently higher, on average, and display a wider distribution with elevated maxima, compared to tropopause height data obtained by radiosonde, suggesting a systematic overestimation.

**Acknowledgement**

The authors would like to acknowledge the National Meteorological Information Centre (NMIC) of CMA, NOAA, Deutscher Wetterdienst (Climate Data Center), U.K Centre for Environmental Data Analysis (CEDA), GRUAN, ECMWF, and the University of Wyoming for continuously collecting and generously providing high–resolution radiosonde data.

**Financial support**

This study jointly supported by the National Natural Science Foundation of China under grants 42205074 and 42127805.

**Competing interests**

The contact author has declared that neither they nor their co–authors have any competing interests



**Data availability**

The radiosonde dataset can be accessed at https://weather.uwyo.edu/upperair/sounding.html and
https://www.ncei.noaa.gov/maps/hourly/. The ERA5 reanalysis dataset can be accessed at
https://datapub.fz-juelich.de/slcs/tropopause/ (Hoffmann and Spang, 2022).

**Author contributions**

JZ conceptualized this study. YG carried out the analysis with comments from other co–authors. YG and JZ wrote the
original manuscript. WW, SZ provided useful suggestions for the study. All authors contributed to the improvement of
paper.

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





**Tables and Figures**

**Table 1.** Tropopause heights determined from WDM, MV and C-F under different SWs or DIs and latitudinal zone. (Unit: km)

|  | Number of sites | WDM |  | 60 m | 100 m | 300 m | 500 m | 700 m | 900 m |
|---|---|---|---|---|---|---|---|---|---|
| **Global Mean** | 387 | 11.17 | **MV** | 11.40 | 11.50 | 11.72 | 11.83 | 11.89 | 11.92 |
|  |  |  | **C-F** | 11.56 | 11.66 | 11.83 | 11.82 | 11.79 | 11.69 |
| **90° N–70° N** | 9 | 8.73 | **MV** | 9.01 | 9.12 | 9.33 | 9.40 | 9.44 | 9.45 |
|  |  |  | **C-F** | 9.17 | 9.26 | 9.36 | 9.31 | 9.24 | 9.16 |
| **70° N–40° N** | 186 | 10.12 | **MV** | 10.35 | 10.44 | 10.63 | 10.71 | 10.75 | 10.77 |
|  |  |  | **C-F** | 10.48 | 10.57 | 10.69 | 10.67 | 10.61 | 10.54 |
| **40° N–20° N** | 110 | 12.86 | **MV** | 13.03 | 13.10 | 13.31 | 13.44 | 13.54 | 13.61 |
|  |  |  | **C-F** | 13.15 | 13.25 | 13.51 | 13.56 | 13.60 | 13.51 |
| **20° N–20° S** | 28 | 15.48 | **MV** | 15.70 | 15.80 | 16.04 | 16.15 | 16.22 | 16.25 |
|  |  |  | **C-F** | 15.87 | 15.98 | 16.15 | 16.15 | 16.09 | 15.97 |
| **20° S–40° S** | 38 | 13.00 | **MV** | 13.18 | 13.27 | 13.54 | 13.71 | 13.83 | 13.92 |
|  |  |  | **C-F** | 13.33 | 13.46 | 13.80 | 13.88 | 13.96 | 13.83 |
| **40° S–70° S** | 15 | 9.48 | **MV** | 9.71 | 9.80 | 10.00 | 10.08 | 10.13 | 10.15 |
|  |  |  | **C-F** | 9.85 | 9.94 | 10.08 | 10.06 | 10.02 | 9.96 |







**Table 2.** Statistical comparison among WDM, MV with SW of 300 m, and C-F with DI of 300 m with ERA5-based. STD: Standard Deviation, CRMSE: Centered Root Mean Squared Error, CC: Correlation Coefficient, Mean difference: the average of this method subtracts ERA5-based. Unit: km.

|  | STD | CRMSE | CC | Mean difference | Absolute mean difference | Median of difference |
|---|---|---|---|---|---|---|
| **ERA5-based** | 2.760 | 0.0 | 1.0 | \ | \ | \ |
| **WDM** | 2.581 | 0.952 | 0.937 | −0.808 | 0.901 | −0.771 |
| **MV 300 m** | 2.584 | 0.912 | 0.942 | −0.278 | 0.486 | −0.094 |
| **C-F 300 m** | 2.635 | 0.871 | 0.948 | −0.181 | 0.424 | −0.100 |





**Figure 1.** Temperature profiles from ground up to 30 km at different stations and release time, in which purple dashed, green dashed and red dashed lines represent tropopause height calculated by WDM, MV (SW equals 500 m) and C-F (DI equals 500 m), respectively. In addition, light blue pentagrams represent CPH. Radiosonde location, date and hour are marked in each subplot.





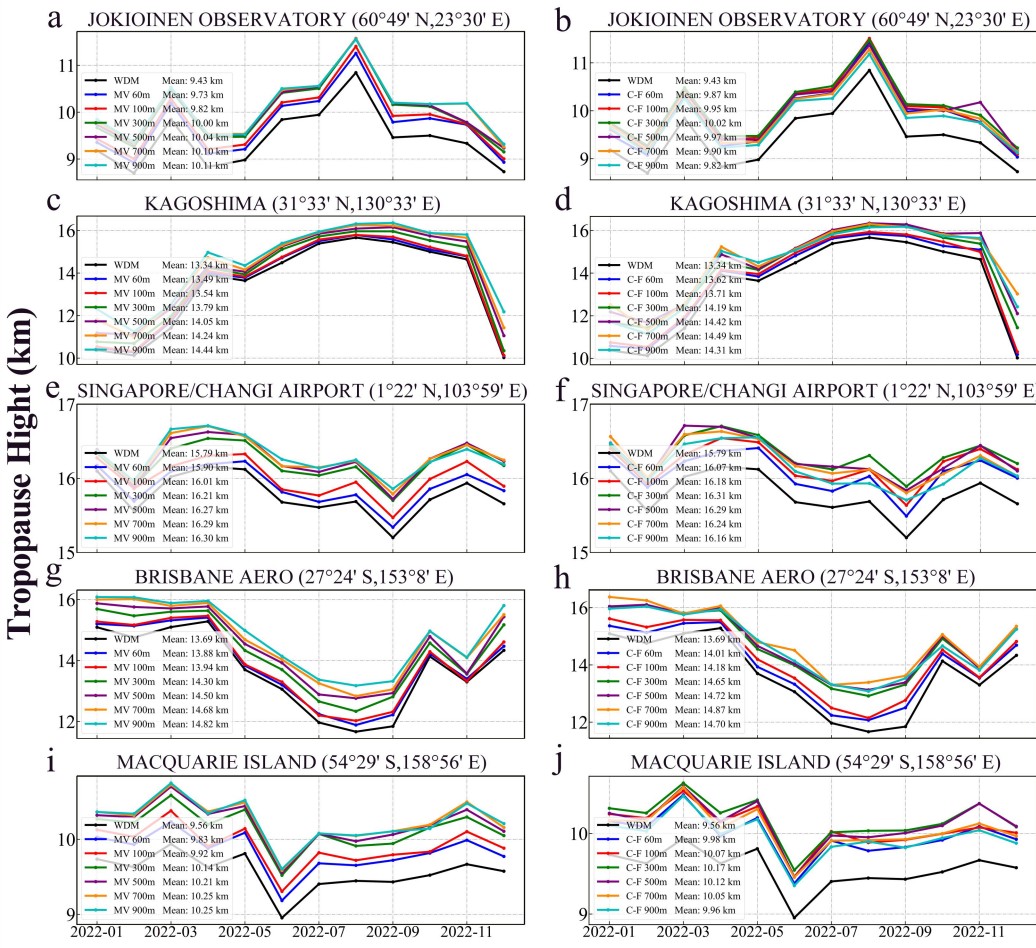


**Figure 2.** Left panel represents the monthly average heights at five stations determined by the MV method under different SWs (60 m, 100 m, 300 m, 500 m, 700 m, 900 m). Similarly, the right panel is for the C-F method under various DIs (60 m, 100 m, 300 m, 500 m, 700 m, 900 m).



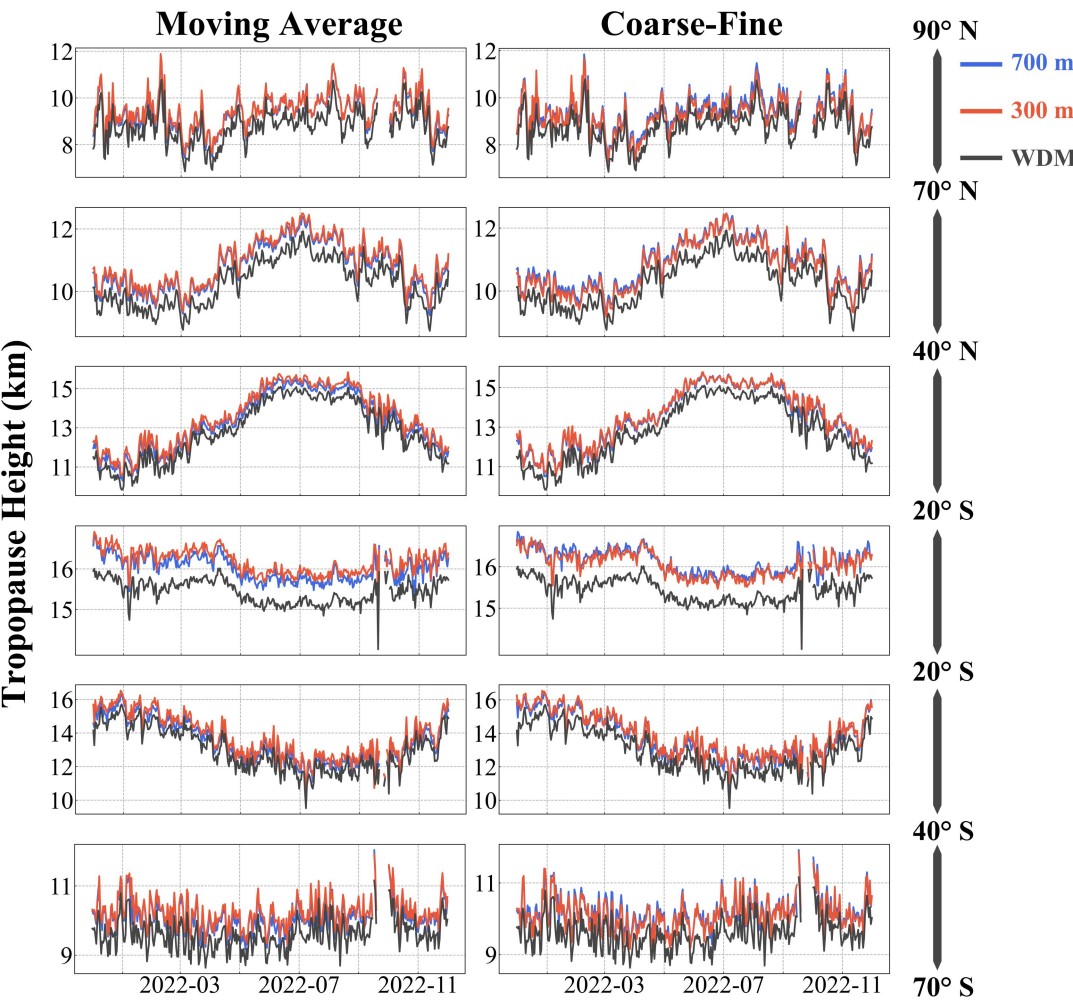

**Figure 3.** Monthly averaged tropopause heights at latitude zones (Station data are absent for the 70° S–90° S region) for the MV (left panel) and C-F (right panel). The black, red, and blue lines represent WDM, MV with SW of 300 m or C-F with DI of 300 m, MV with SW of 700 m or C-F with DI of 700 m.



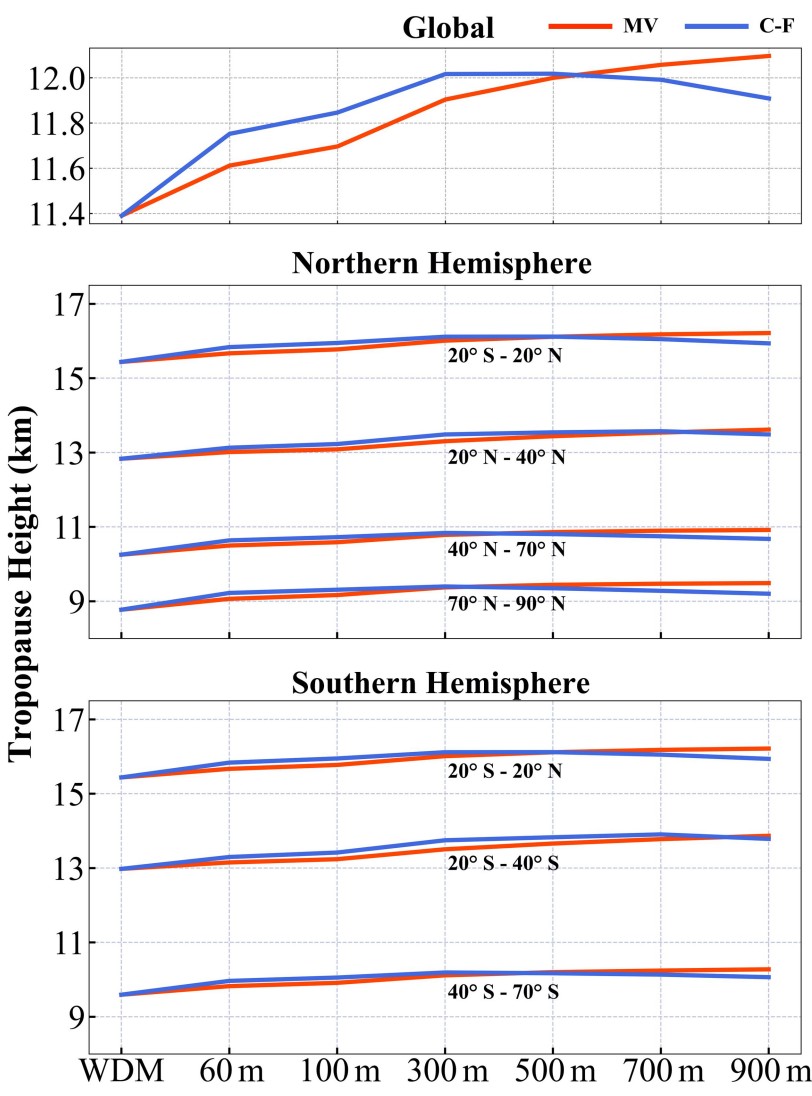

**Figure 4.** Annual variation of tropopause heights across six latitudinal bands and different calculation methods (middle and bottom subplot). The top subplot is for the global mean results. Red and blue lines indicate MV and C-F methods.

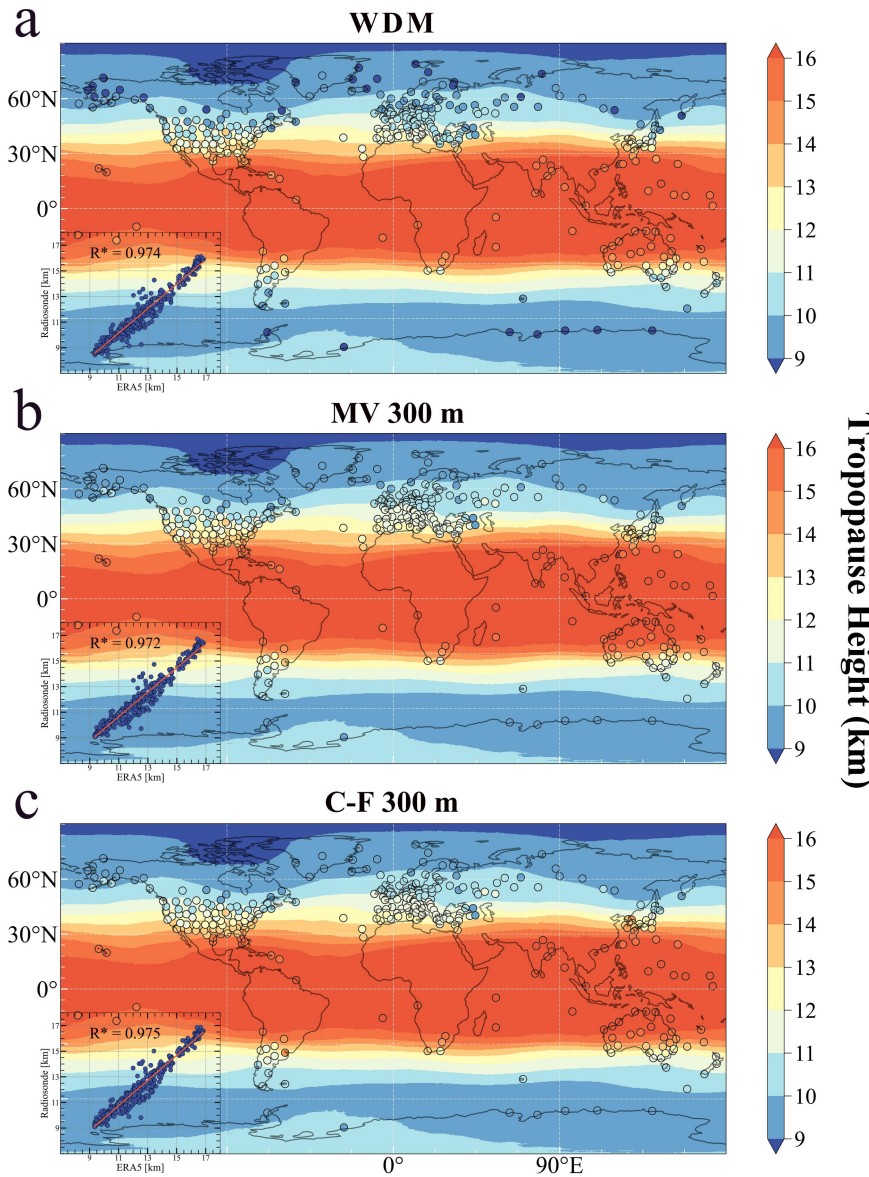

**Figure 5.** Global map of tropopause heights by (a) WDM, (b) MV (SW equals 300 m) and (c) C-F (DI equals 300 m) methods. Each dot represents the annual average data of the first tropopause height at 387 stations in 2022. The base map shows the annual average of the first tropopause height based on the ERA5-based product for 2022. The scatter plots in each map represents the correlation between ERA5 and radiosonde. The correlation coefficient is marked in the lower side, where the star represents confidence level below 0.05.

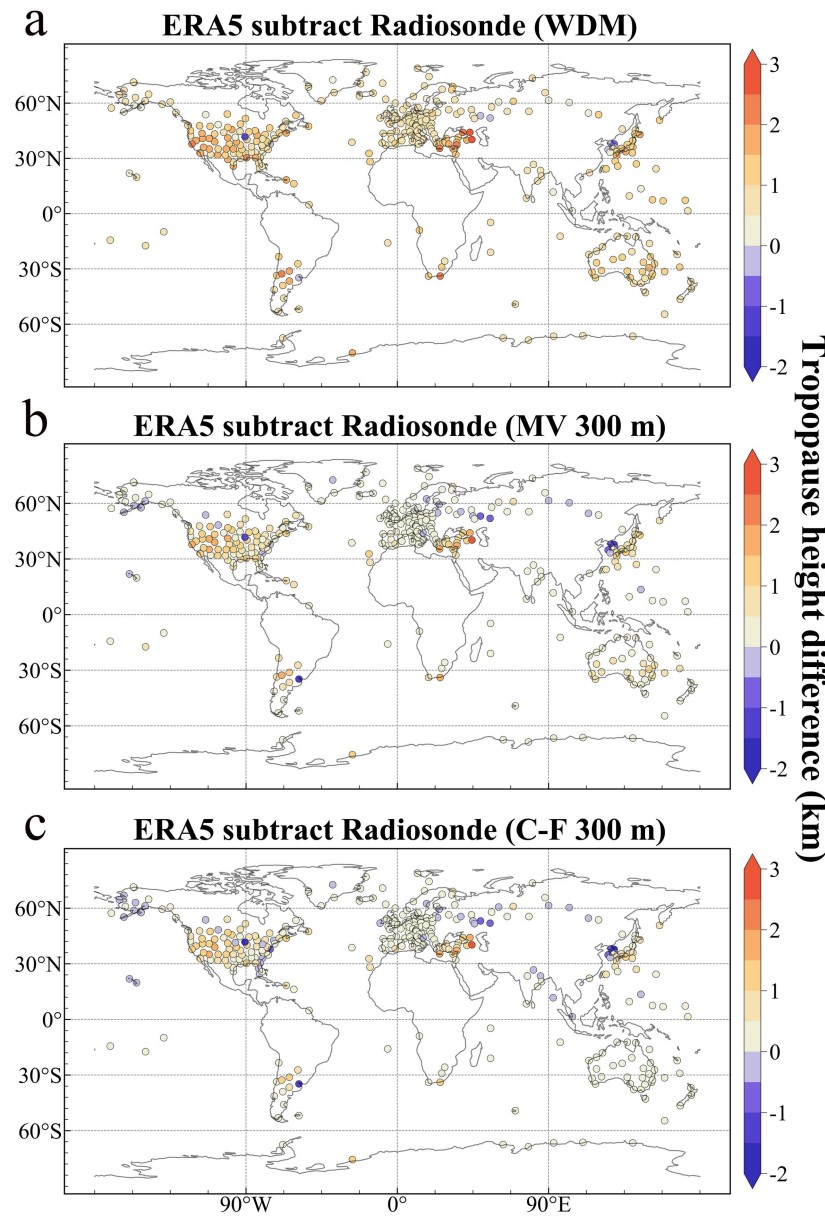

**Figure 6.** The difference between ERA5 tropopause heights and the WDM defined one (a), and the MV with SW of 300 m (b) and the C-F with DI of 300 m (c).





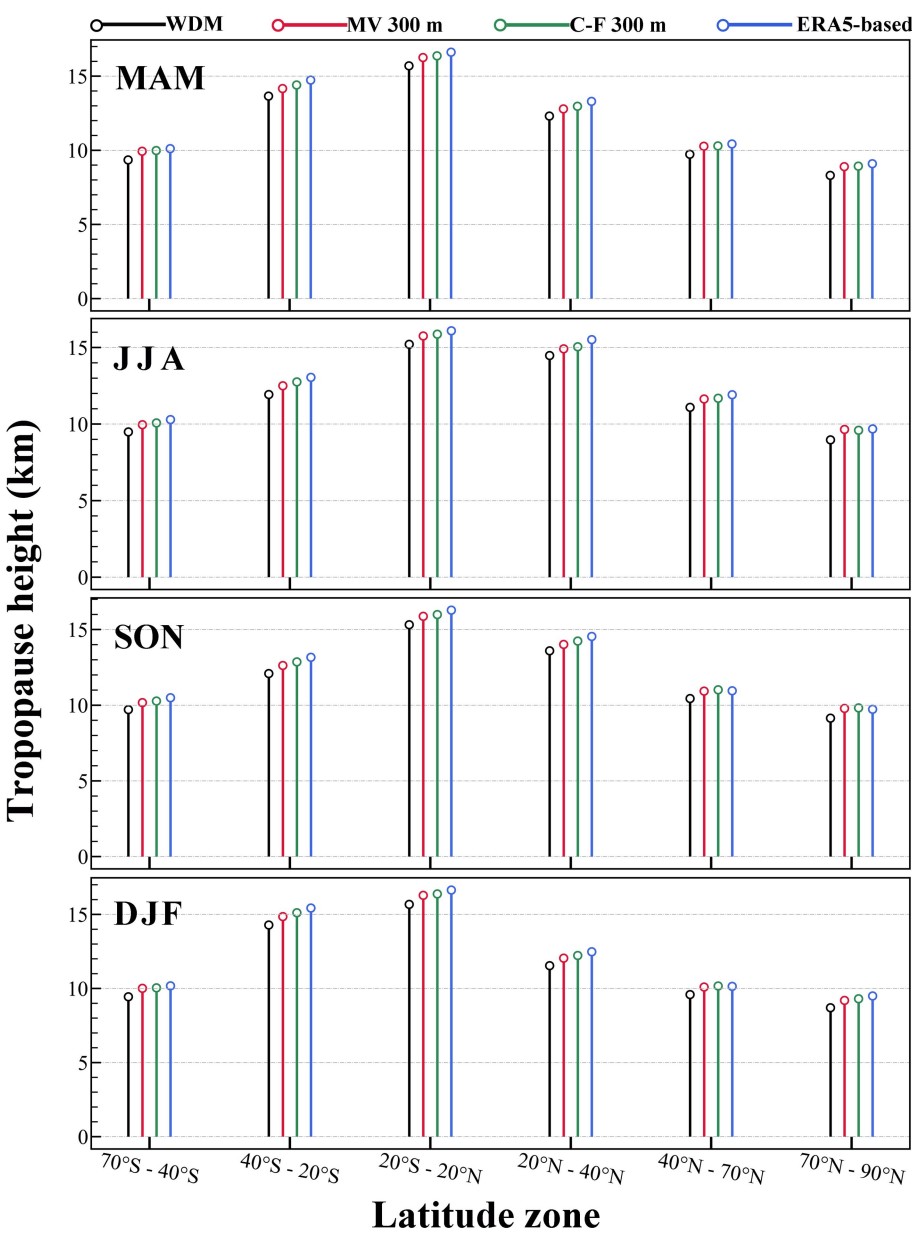

**Figure 7.** A global analysis of average tropopause height during the four seasons of 2022, stratified by six latitude zones. The black line represents the WDM, the red line represents MV (SW equals 300 m), the green line represents the C-F (DI equals 300 m), and the blue line represents ERA5-based product. MAM: March–April–May; JJA: June–July–August; SON: September–October–November; DJF: December–January–February.



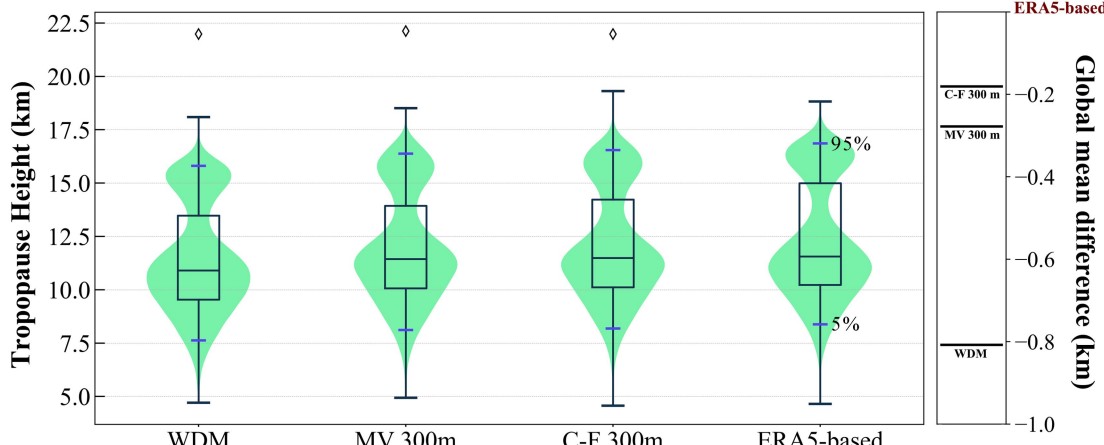

**Figure 8.** Box plots and violin plots for the tropopause height distributions calculated by the WDM, MV (SW equals 300 m), C-F (DI equals 300 m) and ERA5-based. The right-hand figure illustrates the overall discrepancies between the three methods and ERA5-based product.