# Peer review of "The dilemma in identifying WMO-defined tropopause height using high-resolution radiosondes"

_EGUsphere, 2024_

## Community Comment (CC1)

**Response to Referee**

We sincerely appreciate your thoughtful and exhaustive comments and suggestions, which significantly help us to improve the quality of the manuscript.

However, I made a big mistake in my understanding of the tropopause as defined by the World Meteorological Organization, which made my algorithm wrong and on which a large part of the whole article is based. So, we decided to withdraw our manuscript.

Before pulling the manuscript, we would like to express our sincere gratitude to the referee for your exceptionally informative, constructive, and detailed comments, and we would like to answer some of your questions if my algorithm is correct.

**General comment**

This new study by Gou et al. explores the challenges in identifying the tropopause height using high-resolution radiosonde data based on the World Meteorological Organization (WMO) definition. The study highlights that the original WMO approach tends to underestimate the tropopause height in high-resolution datasets due to the presence of thin temperature inversions and gradient discontinuities. To address this issue, the authors propose two alternative methods, the Moving Average (MV) method and the Coarse-Fine (C-F) method, both of which provide more consistent tropopause height estimates compared to ERA5 reanalysis data. It is found that ERA5 systematically overestimates the tropopause height, particularly near the Hadley circulation edges, while the WMO-defined method underestimates it. The C-F method emerges as the most effective in preserving fine-scale structures while filtering out spurious lower-altitude tropopauses. The study concludes that modifications to the WMO method are necessary when using high-resolution radiosonde data for accurate long-term tropopause trend analysis.

The study addresses a crucial problem with practical implications for climate research

and meteorology. The research employs robust statistical techniques, sensitivity analyses, and cross-validation with ERA5, ensuring a high level of scientific rigor. The methodology is well-described, but a deeper discussion on the physical basis of biases and additional validation with independent datasets could enhance transparency. The manuscript is mostly clear and concise. I would like to recommend that the paper be considered for publication, subject to the minor comments listed below.

**Specific comments**

1. l40: It is a little unclear to me what you mean by 'constant emitted temperature'?

Response: The term 'constant' here refers the radiative equilibrium temperature corresponding to infrared emission escaping to space through the atmospheric. This equilibrium temperature must equilibrate with Earth's absorbed solar radiation; deviation from this balance would drive persistent planetary warming or cooling.

Increased atmospheric water vapor enhances optical depth, elevating the effective emission height to colder atmospheric levels. To preserve radiative equilibrium (i.e., maintain constant emission temperature), the system likely modulates tropopause altitude, positioning the emission layer at an elevation that simultaneously compensates for altered optical depth and preserves radiative equilibrium temperature.

2. l106: The dynamical tropopause in the tropics is usually defined by a potential temperature threshold, not by potential vorticity. Please clarify.

Response: This correction provided has been noted and is much appreciated.

This part has been rewritten:

"…The CPT is reliable primarily in the tropics (between 20° S and 20° N) and the dynamic tropopause is only reliable in close proximity to and poleward of the subtropical jets (Xian and Homeyer, 2019)."

3. l116: Lapse rate is defined as temperature difference over height difference. However, for radiosondes this is probably calculated via pressure differences?

Could you please clarify and elaborate?

Response: The radiosonde data employed in this study, obtained from the University of Wyoming, ECMWF, NOAA, contain independent height variables.

4. l119: Reference paper for ERA5 should be cited: Hersbach H, Bell B, Berrisford P, et al. The ERA5 global reanalysis. Q J R Meteorol Soc. 2020; 146: 1999-2049. https://doi.org/10.1002/qj.3803

Response: Amended as suggested.

5. l195: It would be good to know if the high resolution radiosonde data (or a downsampled version of the data) have been assimilated into ERA5? Presumably the data are not independent?

Response: The radiosonde data integrated into ERA5 are based on standard pressure levels with lower resolution, and ERA5 does utilize a downsampled version of the high resolution radiosonde observations (Ingleby, 2017).

Although high-vertical-resolution radiosonde data are part of the assimilation process in established reanalysis data products, it's still provide a good opportunity to quantify uncertainties in the lapse rate tropopause determination from reanalysis data (Hoffmann and Spang, 2022).

Ingleby, B.: An assessment of different radiosonde types 2015/2016, Technical memorandum, https://www.ecmwf.int/en/elibrary/80268-assessment-different-radiosonde-types-20152016, 2017.

Hoffmann, L., and Spang, R.: An assessment of tropopause characteristics of the ERA5 and ERA–Interim meteorological reanalyses, J. Atmos. Chem. Phys., 22, 4019–4046, https://doi.org/10.5194/acp–22–4019–2022, 2022.

Technical corrections

Response: We sincerely appreciate your advice, amended as suggested.

---

## Author Comment (AC1)

**Response to Referee**

**Thank you for identifying this issue during the review. We are very sorry for the time and effort you wasted.**

**I made a big mistake in my understanding of the tropopause as defined by the World Meteorological Organization, which made my algorithm wrong and on which a large part of the whole article is based. So, we decided to adjust the whole article by removing the part of modification to the definition of WMO and adding the long term trend analysis of the tropopause.**

**We would like to express our sincere gratitude to the referee for your exceptionally informative, constructive, and detailed comments, and we would like to answer your comments. We would appreciate it if you could give us your valuable comments for the revised manuscript.**

**Omission of Relevant Prior Research**

1. One related and important general point I would like to make at this stage is that I noticed some glaring omissions of relevant prior work. In particular, given the focus on identification of the tropopause and its characteristics using high-resolution radiosonde data, decades of prior work with such a focus was unexplainably absent. This is especially true for the well-recognized series of early foundational studies by Birner:

   Birner et al., 2002: How sharp is the tropopause at midlatitudes?, Geophys. Res. Lett., 29, doi:10.1029/2002GL015142

   Birner, 2006: Fine-scale structure of the extratropical tropopause, J. Geophys. Res., 111, D04104, doi:10.1029/2005JD006301

   Response: We add Birner (2006) to our new manuscript.

2. These studies and many that follow have demonstrated well that the WMO definition, when applied appropriately, results in reliable definition of the tropopause that is insensitive to profile resolution. There are certainly failure modes

of the WMO definition, but resolution is not one of them (except for resolution that is very coarse, which can of course be ameliorated by interpolation).

Response: Yes, the WMO definition reliably defines the tropopause, the error was due to my mistake, and we have removed the improved section on the WMO definition in the new manuscript.

3. Another topic that I found to be poorly described and referenced relates to the statement at line 45 of the paper. In particular, the abrupt change in tropopause height near the subtropical jet streams is commonly referred to as the tropopause break. There is vast literature on the subject as well as its relation to Hadley Cell (or lack thereof, as several studies have shown that the tropopause break is not coupled to the HC). A search using the term "tropopause break" should turn up these relevant works. Additional studies that explore numerous tropical edge diagnostics to determine changes in the width of the tropics (a la Davis & Rosenlof, 2012, http://dx.doi.org/10.1175/JCL1-D-11-00127.1 and others that follow) will help resolve the HC link argument.

Response: Thanks for the comment. We have added a new reference for a short description.

Line 45: "...The large-scale downwelling in the subtropical Hadley circulation sharply lowers the tropopause, sometimes creating a discontinuity known as the "subtropical tropopause break", which aligns with the STJ (Turhal et al, 2024)."

**Specific comments**

1. The references for the radiosonde data sources appear to be inappropriate. In particular, the University of Wyoming page does not provide access to the full-resolution radiosonde data used in this study, but instead only includes mandatory and significant levels (as in the IGRA archive). The NOAA website referenced appears to only include surface observations and in real-time rather than a historical archive. I believe the most appropriate source of high-resolution radiosonde

observations in the United States is https://www.ncei.noaa.gov/access/metadata/landing-page/bin/iso?id=gov.noaa.ncdc:C01500. There are alternative copies of the data in BUFR format from other sites, which the authors may have used. Because the data I can easily grab from U. Wyoming are on mandatory and significant levels and I do not have time to otherwise download the full-resolution BUFR data, I cannot reproduce Figure 1 from the paper to demonstrate my main point above. Nevertheless, the authors should resolve this data citation issue so that the study could be easily replicated by any reader.

Response: We have elaborated this point in the revised manuscript. URL links are available in the Data availability section of the manuscript.

Line 115: "…Following Guo et al. (2021) and Zhang et al. (2022), we utilized a high-vertical-resolution radiosonde (HVRRS) dataset spanning 2000 to 2023 (24 years), compiled from multiple sources including the China Meteorological Administration (CMA), the National Oceanic and Atmospheric Administration (NOAA) of the United States, the German Deutscher Wetterdienst (Climate Data Center), the Centre for Environmental Data Analysis (CEDA) of the United Kingdom, the Global Climate Observing System (GCOS) Reference Upper-Air Network (GRUAN), and the University of Wyoming."

2. Lines 74-75: I am not sure what this sentence means or implies.

Response: Original quote: "…We found that simulations with near-neutral convective lapse rate, which corresponds to cold climates in the moist GCM, have a poorly defined tropopause when applying the WMO definition. To circumvent this issue, we use a different definition of the tropopause height based on the meridional circulation structure."

And we have removed this paragraph in the new manuscript.

Levine, X. J., and Schneider, T.: Baroclinic Eddies and the Extent of the Hadley Circulation: An Idealized

GCM Study, J. Atmos. Sci., 72, 2744–2761, https://doi.org/10.1175/JAS-D-14-0152.1, 2015.

**3.** Line 85: "retrieve" should be "retrieval"

Response: Amended as suggested.

---

## Author Comment (AC2)

**Response to Referee**

**We sincerely appreciate your thoughtful and exhaustive comments and suggestions, which significantly help us to improve the quality of the manuscript.**

**However, I made a big mistake in my understanding of the tropopause as defined by the World Meteorological Organization, which made my algorithm wrong and on which a large part of the whole article is based. So, we will adjust the whole article by removing the part of modification to the definition of WMO and adding the long term trend analysis of the tropopause.**

**We would like to express our sincere gratitude to the referee for your exceptionally informative, constructive, and detailed comments, and we would like to answer some of your questions if my algorithm is correct. We would appreciate it if you could give us your valuable comments for the revised manuscript.**

**General comment**

This new study by Gou et al. explores the challenges in identifying the tropopause height using high-resolution radiosonde data based on the World Meteorological Organization (WMO) definition. The study highlights that the original WMO approach tends to underestimate the tropopause height in high-resolution datasets due to the presence of thin temperature inversions and gradient discontinuities. To address this issue, the authors propose two alternative methods, the Moving Average (MV) method and the Coarse-Fine (C-F) method, both of which provide more consistent tropopause height estimates compared to ERA5 reanalysis data. It is found that ERA5 systematically overestimates the tropopause height, particularly near the Hadley circulation edges, while the WMO-defined method underestimates it. The C-F method emerges as the most effective in preserving fine-scale structures while filtering out spurious lower-altitude tropopauses. The study concludes that modifications to the WMO method are necessary when using high-resolution radiosonde data for accurate long-term tropopause trend analysis.

The study addresses a crucial problem with practical implications for climate research and meteorology. The research employs robust statistical techniques, sensitivity analyses, and cross-validation with ERA5, ensuring a high level of scientific rigor. The methodology is well-described, but a deeper discussion on the physical basis of biases and additional validation with independent datasets could enhance transparency. The manuscript is mostly clear and concise. I would like to recommend that the paper be considered for publication, subject to the minor comments listed below.

**Specific comments**

1. l40: It is a little unclear to me what you mean by 'constant emitted temperature'?

Response: The term 'constant' here refers the radiative equilibrium temperature corresponding to infrared emission escaping to space through the atmospheric. This equilibrium temperature must equilibrate with Earth's absorbed solar radiation; deviation from this balance would drive persistent planetary warming or cooling.

Increased atmospheric water vapor enhances optical depth, elevating the effective emission height to colder atmospheric levels. To preserve radiative equilibrium (i.e., maintain constant emission temperature), the system likely modulates tropopause altitude, positioning the emission layer at an elevation that simultaneously compensates for altered optical depth and preserves radiative equilibrium temperature.

2. l106: The dynamical tropopause in the tropics is usually defined by a potential temperature threshold, not by potential vorticity. Please clarify.

Response: This correction provided has been noted and is much appreciated. We want to show here that the dynamical tropopause is not available in that region.
This part has been rewritten:
Line 51: "…the dynamic tropopause (WMO, 1985; Hoinka, 1998), typically defined by potential vorticity (PV) thresholds of 1.5–4 potential vorticity unit (PVU) (Turhal et al., 2024), is less reliable in regions of low absolute potential vorticity, such as the tropics, and sometimes in mid-latitudes where strong anticyclonic flow prevails (Hoerling et al.,

1991)"

3. l116: Lapse rate is defined as temperature difference over height difference. However, for radiosondes this is probably calculated via pressure differences? Could you please clarify and elaborate?

Response: The radiosonde data employed in this study, obtained from the University of Wyoming, ECMWF, NOAA, contain independent height variables.

4. l136: Please clarify the shape of the moving window. Presumably a boxcar/tophat function was used?

Response: It's a boxcar function. It was a sliding average, which has now been removed from revised manuscript.

5. l119: Reference paper for ERA5 should be cited: Hersbach H, Bell B, Berrisford P, et al. The ERA5 global reanalysis. Q J R Meteorol Soc. 2020; 146: 1999-2049. https://doi.org/10.1002/qj.3803

Response: Amended as suggested.

6. l195: It would be good to know if the high resolution radiosonde data (or a downsampled version of the data) have been assimilated into ERA5? Presumably the data are not independent?

Response: The radiosonde data integrated into ERA5 are based on standard pressure levels with lower resolution, and ERA5 does utilize a downsampled version of the high resolution radiosonde observations (Ingleby, 2017).

Although high-vertical-resolution radiosonde data are part of the assimilation process in established reanalysis data products, it's still provide a good opportunity to quantify uncertainties in the lapse rate tropopause determination from reanalysis data (Hoffmann and Spang, 2022).

Ingleby, B.: An assessment of different radiosonde types 2015/2016, Technical memorandum, https://www.ecmwf.int/en/elibrary/80268-assessment-different-radiosonde-types-20152016, 2017.

Hoffmann, L., and Spang, R.: An assessment of tropopause characteristics of the ERA5 and ERA–Interim meteorological reanalyses, J. Atmos. Chem. Phys., 22, 4019–4046, https://doi.org/10.5194/acp–22–4019–2022, 2022.

7.  Fig. 6: These maps suggest that the statistical differences are due to a few individual outliers, probably related to inversions, rather than a bias across all profiles.

Response: The data used for this graph is wrong, but you can see the same type of comparison graph in our revised manuscript, where the prominence of individual cases is allowed in the statistics, and it will be diluted by a large amount of overall error, even though in the new manuscript their differences are small.

Technical corrections

Response: We sincerely appreciate your advice, amended as suggested.

---

## Author Comment (AC3)

**Response to Referee**

**We sincerely appreciate your thoughtful and exhaustive comments and suggestions, which significantly help us to improve the quality of the manuscript.**

**However, I made a big mistake in my understanding of the tropopause as defined by the World Meteorological Organization, which made my algorithm wrong and on which a large part of the whole article is based. So, we will adjust the whole article by removing the part of modification to the definition of WMO and adding the long term trend analysis of the tropopause.**

**We would like to express our sincere gratitude to the referee for your exceptionally informative, constructive, and detailed comments. We would appreciate it if you could give us your valuable comments for the revised manuscript.**

**General comment**

I have read this manuscript, and unfortunately, it falls short in terms of quality and presentation. What is worse, it seems to be built on a set of misunderstandings on what tropopause is, apparently, a lack of knowledge of the existing literature and research performed on it. The lack of discussion in the Introduction of the existing work in the field and even the doubtful use of some papers to support some statements (or not citing them) is another issue of concern here. I develop all this below, and because of it, I consider that this manuscript must be rejected for publication, not in this journal, but in any other in its current form. I do not like to be so categorical, but this manuscript presents serious flaws in this case.

First, as evident from the work presented, the title of the manuscript is grandiloquent, overstating, and not representative of the work presented here, which is, at best, a small study on how different statistics give different results. The fact that is applied to the tropopause is a minor detail. It does not exist any dilemma in the scientific community on what it is the tropopause or how to define it, and high-resolution radiosondes,

mentioned in the title as if they were a key part of the work, are not compared against other radiosondes, so it is not possible to establish here anything from them. Also, it has been proved that in some cases, the vertical resolution of the radiosonde data compared to IGRA, which they mention here, does not impact the final statistics on tropopause. At least this previous work should be cited in the text: Antuña et al. (2006) https://doi.org/10.5194/angeo-24-2445-2006

The authors try to propose better tropopause "definitions" that accommodate the high-resolution radiosonde data. However, they never study if they are seeing tropospheric and stratospheric air masses, and their reasoning to support their criteria as valid is not backed by anything and obviates all the existing literature on the complex structure of the tropopause, their inversion layer, multiples tropopauses, latitudinal transport of air masses, and regional characteristics. Even their split in different regions does not follow the usual practice in the community, which makes any potential intercomparison hard.

Some parts of the manuscript contain incorrect statements, and the presentation presents flaws in some parts. These problems with the quality of the manuscript submitted undermine confidence in the quality of the work presented here.

Response: These criticisms were invaluable, and I recognized my mistakes and reworked the entire manuscript, removing the section on improvements in the definition of WMO and adding a study of long-term trends in tropopause heights.

**Specific comments**

1.  Line 32: It is hard to understand the selection of the work by Xian and Homeyer (2019) to establish in a paper what tropopause is in the atmosphere. With all the due respect to their work, their study is one more in a large existing literature about the tropopause, and is cherry-picked here. One would reasonably expect that the first paper to be cited to explain what is the tropopause is a review paper, a seminal work or book, such as the definition in the Encyclopedia of Atmospheric Sciences,

or a paper such as Fueglistaler et al. (2009) or Gettelman et al. (2011).

Response: Amended as suggested.

Line 31: "…The tropopause, marking the boundary between the turbulent troposphere and the stably stratified stratosphere, is a "gate" for exchange of energy, air masses, water vapor and so-called very short lived substances (Fueglistaler, 2009)."

2.  Line 45: "defining" the tropopause does not lack a universal approach. The tropopause has a definition given by the WMO. What can be different is the separation between a tropospheric and a stratospheric regime. Obviously, different characteristics of the troposphere and stratosphere produce a transition between both layers that is different from the tropopause thermal definition, but this does not mean that the tropopause does not have a definition.

Response: This part has been removed.

3.  It is important that the authors make a clear difference here between the characteristics that define the layer that separates the troposphere and the stratosphere regimes and the transition layer itself. This can be done by clarifying that the tropopause has a definition, but many "criteria" can be used to separate the tropospheric and stratospheric characteristics. The dynamic criterion (usually based on Potential Vorticity fields) does not fail at midlatitudes but in the tropics, where potential vorticity isolines get vertical. Actually, the authors make the point correctly in line 107. If they are going to talk about the different criteria in a manuscript that intends to deal with the determination of the tropopause, one would also expect a complete overview of it, mentioning the e90 by Prather et al. and the different PV values tested by Hoinka.

Response: Amended as suggested. We have added citations and a short note.

Line 51: "…and the dynamic tropopause (WMO, 1985; Hoinka, 1998), typically

defined by potential vorticity (PV) thresholds of 1.5–4 potential vorticity unit (PVU) (Turhal et al., 2024), is less reliable in regions of low absolute potential vorticity, such as the tropics, and sometimes in mid-latitudes where strong anticyclonic flow prevails (Hoerling et al., 1991).”

4. Line 47: It is important to include citations to the different criteria to determine the tropopause for those unfamiliar with the topic. One of the best for the CPT is the paper by Gettelman et al. (2000), which began to make popular the topic. For the dynamical tropopause, the combination of the WMO (1985) technical report and the paper by Hoinka (1998) give the necessary information.

Response: For the CPT we have added a reference: Highwood and Hoskins, 1998. For the dynamical tropopause, amended as suggested.

Line 49: “…The cold point tropopause (CPT) get the tropopause via the minimum temperature in the vertical temperature profile, which is unsuitable for extratropical regions (Highwood and Hoskins, 1998), and the dynamic tropopause (WMO, 1985; Hoinka, 1998), typically defined by potential vorticity (PV) thresholds of 1.5–4 potential vorticity unit (PVU) (Turhal et al., 2024), is less reliable in regions of low absolute potential vorticity, such as the tropics, and sometimes in mid-latitudes where strong anticyclonic flow prevails (Hoerling et al., 1991).”

5. It is also important to note that the WMO definition (as anyone thermal) fails in polar regions, because of the low vertical gradient that exists in those regions during parts of the year, contrary to what the authors state in lines 49-50. And line 49 should read "The tropopause definition (WMO, 1957), which is thermodynamic..."

Response: Amended as suggested.

Line 54: “However, the tropopause definition (WMO, 1957), which is thermodynamic, proposed by the World Meteorological Organization offers a more robust global

approach (though it may occasionally fail in polar regions), providing reliable TH estimates from various datasets (Hoffmann and Spang, 2022)."

6. Line 51: The upward trend in the tropopause is not recent. The first report on it, already cited by the authors, is from 2003, and the existing data for this trend is from radiosonde data and goes back to 1958 (Seidel and Randel, 2006). I would cite here one of my papers, contemporaneous to the one of Seidel and Randel, Añel et al. (2006).

Response: Amended as suggested.

Line 70: "…More and more evidence suggests an upward trend in TH under a changing climate (Santer et al., 2003; Sausen and Santer, 2003; Seidel and Randel, 2006; Añel et al., 2006)."

7. Lines 54-57: It is hard to understand how this manuscript does not include a citation to the paper by Meng et al. (2021) https://doi.org/10.1126/sciadv.abi8065, which is one of the most recent discussing the tropopause rise from radiosondes.

Response: Amended as suggested.

Line 74: "…Meny et al. (2021) reported a TH increase of 50–60 m/decade (2001–2020) in the Northern Hemisphere based on radiosonde data."

8. Line 58: You should cite here the papers for IGRA: Durre et al. (2006, 2018), which you already include in the list of references.

Response: Amended as suggested.

Line 77: "…It is worthwhile to note that these radiosonde-related studies use data from the Integrated Global Radiosonde Archive (IGRA), a global dataset with coarse vertical resolution (approximately 300 to 400 m) (Durre et al., 2006, 2018),…"

9. Line 63: It is unclear what the authors want to say here. A couple of sentences before, they say that radiosonde data (IGRA) suffers from vertical resolution problems; now, here, they say that radiosonde data have greater vertical resolution. Do you refer now here to "raw" radiosonde data and not the soundings included in IGRA? Please clarify.

Response: Amended as suggested.

10. Line 70: "tropopause also exits as inversions". It is unclear what you want to say or imply with this sentence fragment. Please clarify.

Response: Amended as suggested. We delete it.

11. Line 78: it is unnecessary to clarify that ERA5 is the sucessor to ERA-Interim, please, delete it.

Response: Amended as suggested.

12. Line 94: quite self-serving, but if you talk about studying the tropopause structure from radiosonde data, you are probably missing the only study that is important to cite (jointly with the Birner et al. (2006) paper on the TIL), the one that studies its global structure until the third tropopause level: Añel et al. (2008) Climatological features of global multiple tropopause events (https://doi.org/10.1029/2007JD009697).

Response: Amended as suggested.

Line 106: "…tropopause structure (Birner, 2006; Seidel and Randel, 2006; Añel et al., 2008; Sunilkumar et al., 2017)…"

13. Line 100: Using a previously published paper to cite three different data sources is far from being the best practice. Instead, you should provide the exact data repositories for the datasets that you use. Following the better current standards on data citation, at least they should have a DOI. It would be great if they had it.

Response: Acquiring globally representative high-resolution radiosonde data necessitates integration of multiple observational sources, as no single dataset provides complete spatial coverage. URL links are available in the Data availability section of the manuscript.

Line 115: "…Following Guo et al. (2021) and Zhang et al. (2022), we utilized a high-vertical-resolution radiosonde (HVRRS) dataset spanning 2000 to 2023 (24 years), compiled from multiple sources including the China Meteorological Administration (CMA), the National Oceanic and Atmospheric Administration (NOAA) of the United States, the German Deutscher Wetterdienst (Climate Data Center), the Centre for Environmental Data Analysis (CEDA) of the United Kingdom, the Global Climate Observing System (GCOS) Reference Upper-Air Network (GRUAN), and the University of Wyoming."

14. Line 105: again, no several methods exist to define the tropopause. Maybe to separate the troposphere from the stratosphere.

Response: This part has been removed.

15. It is Ertel's Potential Vorticity, not Rossby-Ertel. Rossby's formalism does not contain the features later developed by Ertel (conservation) that make the EPV a tracer that can separate stratosphere and troposphere air masses.

Response: Thanks for the correction. This section was not included after the revised manuscript.

16. Lines 111 and 114: I am being picky here, but the WMO definition does not contain the quotation marks used by the authors here, but the terms "first tropopause" and "second tropopause" are given in italics. I think that the right way to do it would be to put the full definition between the quotation marks and not only these two terms.

Response: Amended as suggested.

17. Lines 150-152: I understand the argumentation by the authors; however, the statement "are likely to be more consistent with previous research." what do you mean by this? With what previous research? Why? This sounds like a willingness more than a scientific fact, and no previous research is cited to support such a statement. Then you say, "and understanding of the tropopause." This statement is baseless. Here, in these soundings, the authors do not check what the troposphere or the stratosphere is. For example, in Fig. 1a,b, have you checked if the air mass between the WDM and the MV or C-F is of stratospheric or tropospheric origin? Guessing that the WDM is not a valid tropopause is not enough; if you are stating it, you should prove it, for example, with PV and ozone fields. This station is located around the subtropical jet stream, a region prone to the occurrence of multiple tropopauses (see Añel et al. 2008), and where the tropospheric and stratospheric air masses mix along several km in the vertical. You can not simply state that the WDM is a not valid tropopause when, for January, the tropopause can appear in the region of this sounding at levels so low as 200 hPa (see Añel et al. (2008) again, e.g., Fig.4), and in a latitude where the latitudinal entrainment of mid-latitudes stratospheric air under the tropical tropopause makes hard to define the change between the tropospheric and stratospheric air masses. See Wang and Polvani (2011) and Añel et al. (2012).

Response: Thanks for the criticism, this part of my argument is insufficiently supported. The revised manuscript no longer contains that image and its interpretation of its contents.